# Chirality control of a single carbene molecule by tip-induced van der Waals interactions

Yunjun Cao [1], Joel Mieres-Perez[2,3], Julien Frederic Rowen [4], Elsa Sanchez-Garcia[2,3], Wolfram Sander [4] & Karina Morgenstern [1] ✉

Non-covalent interactions such as van der Waals interactions and hydrogen bonds are crucial for the chiral induction and control of molecules, but it remains difficult to study them at the single-molecule level. Here, we report a carbene molecule on a copper surface as a prototype of an anchored molecule with a facile chirality change. We examine the influence of the attractive van der Waals interactions on the chirality change by regulating the tip-molecule distance, resulting in an excess of a carbene enantiomer. Our model study provides insight into the change of molecular chirality controlled by van der Waals interactions, which is fundamental for understanding the mechanisms of chiral induction and amplification.

Non-covalent interactions, such as van der Waals (vdW) interactions and hydrogen bonding, play a predominant role in chiral recognition, induction, and amplification[1,2]. By non-covalent interactions, molecules from a racemic mixture are recognized in the same chirality to form local homochirality[3,4]. To achieve global homochirality, chiral seeds like chiral guest molecules, using the "sergeants-and-soldiers" principle[5–7], or a small enantiomeric excess in reactants, the "majority-rules" effect[8–10], are introduced to induce and amplify chirality. A first and elementary step in chiral amplification is chiral induction. It is initiated by non-covalent interactions, interactions that are omnipresent between adjacent molecules[11]. However, addressing the role of these interactions in chiral induction at the fundamental level is still a grand challenge.

Scanning probe techniques, such as atomic force microscopy (AFM) and scanning tunneling microscopy (STM), offer the possibility to characterize local interactions of individual molecules. AFM is generally very powerful in terms of interaction measurements, but it requires strong interactions for the manipulation of a molecule[12,13]. In contrast, weak non-covalent interactions, like vdW interactions between adjacent molecules, are sufficient in chiral induction to make that a molecule prefers one enantiomer in favor of the other via subtly altering the potential well (Interaction, Fig. 1a). Meanwhile, the energy barrier of the chirality change is overcome by temperature (Activation, Fig. 1a). To mimic chiral induction in a molecular assembly, we propose a strategy where a STM tip provides the vdW interactions to alter the

potential well, representing the molecular vdW interactions (Fig. 1a,b). In molecular assemblies, the energy barrier between the two chiral enantiomers is supposed to be thermally driven. It is replaced here by the injection of inelastic electrons into the molecules. Such a strategy facilitates addressing the influence of weak vdW interactions on the dynamics of chirality changes.

To address the influence of vdW interactions in chiral induction, a prerequisite is to switch a molecule between its two enantiomeric states in a controllable, precise, and facile fashion. Photo-induced and thermally-initiated processes in molecular assemblies cannot lead to an enantiomeric excess since these processes are not unidirectional[14,15]. For tip-induced chirality changes of individual molecules by inelastic electron tunneling (IET), the conversion has to compete with side processes like rotation, diffusion, or even dissociation[16–19]. Moreover, most chirality changes require that parts of the molecule are lifted from the surface, corresponding to energy barriers of hundreds of milli-electron volts (meV)[14–17]. Such high barriers make it hard to examine the contribution of weak intermolecular interactions on chirality changes, especially the weakest of them, the vdW interactions, which is typically a few meV per bond[20]. Overall, chirality changes of individual molecules were achieved by photons[14], heat[15], and electrons[16,17,19]. However, the key to chiral induction, the influence of vdW interactions on this process, has not been addressed yet.

In this article, we present diphenylcarbene (DPC) as a prototypical anchored molecule, going through a facile chirality change on a

[1]Physical Chemistry I, Ruhr-Universität Bochum, Universitätsstr. 150, D-44801 Bochum, Germany. [2]Computational Bioengineering, Technical University Dortmund, Emil-Figge-Str. 66, 44227 Dortmund, Germany. [3]Computational Biochemistry, Universität Duisburg-Essen, Universitätsstr. 2, D-45141 Essen, Germany. [4]Organic Chemistry II, Ruhr-Universität Bochum, Universitätsstr. 150, D-44801 Bochum, Germany. ✉e-mail: karina.morgenstern@rub.de

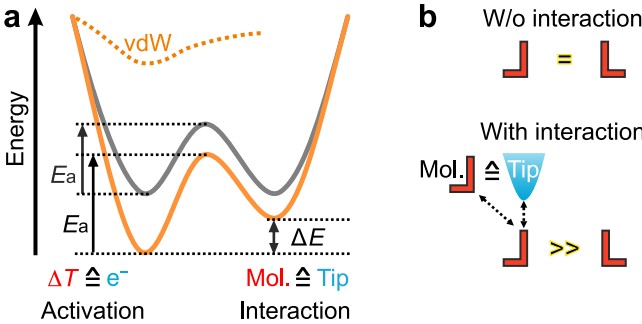

**Fig. 1 | Schematics of molecular chirality change induced by adjacent molecule or tip. a** Schematic potential wells for chirality changes. Solid gray curve represents the native potential well for the chirality change of a molecule (Mol.) without interaction, dashed orange curve represents the net potential induced by an adjacent molecule or tip, and solid orange curve represents the total potential well in the presence of an adjacent molecule or tip. $E_a$ marks the activation energy of a chirality change. e⁻ and $\Delta T$ mark inelastic electrons injected by a STM tip and temperature, respectively. **b** Schematic chirality changes of a single molecule (red) without (w/o) interaction and with interaction; dashed arrows mark molecule-molecule interactions and tip-molecule interactions.

copper surface without side processes. The two nonplanar phenyl rings of DPC make it helically chiral in gas phase, while its rotatable C–C–C bonds facilitate the chirality change. Upon adsorption on a copper surface, a chirality change of DPC is achieved by manipulating one of its phenyl rings with inelastically tunneling electrons. In contrast to weakly adsorbed molecules on surfaces[16,17], the carbene center serves as an anchor[21] that is covalently bonded to the copper surface, suppressing any side processes upon excitation. By controlling the vdW interactions with the STM tip, we are able to obtain an excess of one of the enantiomers. The role of the vdW interactions in this process is further validated by a numerical analysis based on a Lennard-Jones potential. Our study takes advantage of inelastic electrons and tip-induced vdW interactions to mimic a chiral induction in a molecular assembly. The subtle influence of vdW interactions is used to gain control over the chirality, providing a molecular-level understanding of chiral induction and amplification.

## Results and discussion

### Surface anchoring by carbene center

Recently, a highly reactive carbene, fluorenylidene, with a planar, rigid configuration was characterized on a Ag(111) surface[22]. The strong interaction of fluorenylidene with the metal surface via its carbene center identifies such centers as suitable candidates for a suppression of unintended processes upon excitation. Here, a carbene with a flexible and three-dimensional configuration, DPC, is characterized on a Cu(111) surface. The reactivity of both carbenes is very similar[23,24]. Because of its three-dimensional structure, DPC is chiral, whereas the planar fluorenylidene is achiral. The conformational flexibility of DPC originates from its rotatable C–C–C bonds, which is corroborated by a low energy barrier of the conversion between its two enantiomeric states in gas phase (Supplementary Fig. 1).

On Cu(111), DPC is formed on the copper surface by the loss of $N_2$ from the diphenyldiazomethane (DPDM) precursor, taking advantage of the copper-catalyzed dissociation of diazo groups through the Doyle-Kirmse reaction (Fig. 2a)[25,26]. The isolated DPC molecules are randomly distributed on Cu(111) (Fig. 2b). Their chiral nature is revealed by curved shapes in high-resolution STM images. The two enantiomers cannot be superimposed with each other by rotation or translation (Fig. 2c, d). The two protrusions of each enantiomer (1 and 2, Fig. 2c, d) show a difference of 6 pm in apparent height (Fig. 2e). Adjacent depressions (squares in Fig. 2c, d) are related to a charge transfer from Cu(111) to the carbene center, analogous to that of

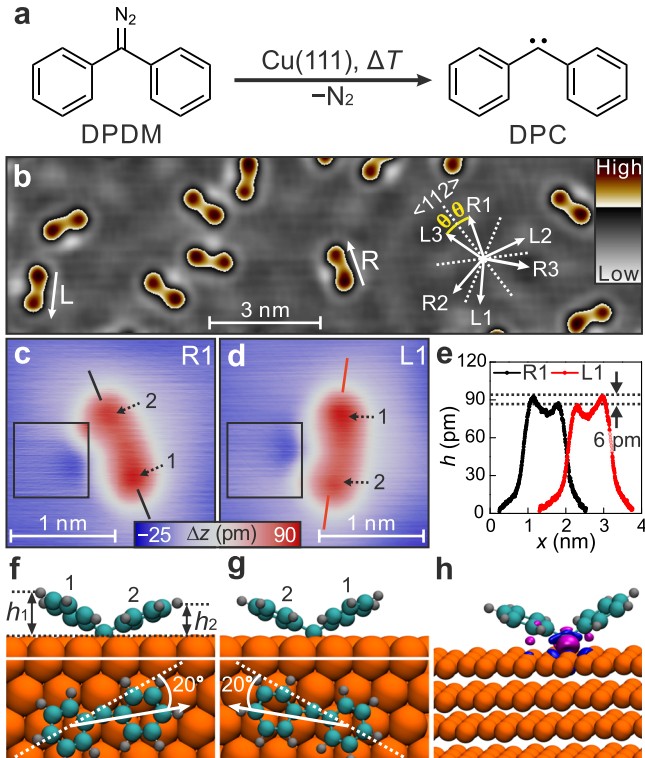

**Fig. 2 | Carbene adsorption on Cu(111). a** Scheme of the reaction of the precursors DPDM to DPC on Cu(111). **b** STM overview image of DPC formed by depositing DPDM on Cu(111) at 250 K. R and L mark one right-handed and one left-handed DPC enantiomer, respectively. The white arrows mark the directions of R-type and L-type enantiomers along their main molecular axes, where $\theta = \pm (16 \pm 2)°$ are their inclination angles with respect to the <112> surface directions. The STM image was processed by a Laplace-filter (for details see Methods Section) to enhance the contrast. **c,d** Magnified images of one R-type (**c**) and one L-type (**d**) enantiomers. The contrast is enhanced in the black squares to make the faint depressions visible. 1 and 2 mark the brighter and less-bright protrusions. Scanning parameters: (**b**) $V_b = 50$ mV, $I_t = 5$ pA, (**c, d**) $V_b = 10$ mV, $I_t = 1$ nA. **e** Height profiles along the lines in (**c**) and (**d**) in corresponding colors. **f, g** The optimized structures of the two DPC enantiomers on Cu(111) in side view (upper panel) and top view (lower panel). Orange spheres: copper; cyan spheres: carbon; gray spheres: hydrogen. $h_1$ and $h_2$ in (**f**) mark the distances between the surface plane and the hydrogen atoms furthest away from the surface for each ring (1 and 2). **h** Charge density difference between DPC/Cu(111) and the individual components, DPC and Cu(111), at the same atomic positions as in the adsorbed system. Blue and pink regions represent charge depletion and charge accumulation, respectively. The isosurface value is $4 \cdot 10^{-3}$ e⁻ bohr⁻³.

fluorenylidene on a Ag(111) surface[22]. It indicates that the carbene carbon atoms are situated at the sides of the depressions.

The two enantiomers further differ in their orientations, by $\theta = \pm (16 \pm 2)°$ with respect to the <112> directions of the surface (Fig. 2b). We assign the enantiomers in pairs based on the positions of the anchoring points (depressions) with respect to the two phenyl rings (protrusions), i.e., R$n$⟷L$n$ ($n = 1$, 2, and 3, Supplementary Fig. 2). As discussed below, the anchor to the surface suppresses any diffusion or flip of the molecules upon excitation.

To elucidate the interaction between DPC and Cu(111), we performed density functional theory (DFT) calculations. Our study reveals a configuration of DPC with out-of-plane tilting of the two phenyl rings with respect to the surface (Fig. 2f). The calculated adsorption heights of the two phenyl rings at $h_1 = 0.44$ (0.42) nm and $h_2 = 0.36$ (0.38) nm (top section in Fig. 2f, g), defined by the highest hydrogen atoms of each ring above the surface plane, corroborate the different contrast of the two protrusions of DPC in the STM image

(Fig. 2c, d). A closer inspection reveals that the lower phenyl rings (2) adsorb close to hollow sites of Cu(111) with the higher phenyl rings (1) to bridge sites (Fig. 2f, g). The different adsorption sites of the two rings cause different ring-surface interactions and thus adsorption heights. The main molecular axes of the two calculated DPC enantiomers are rotated by $\pm 20°$ with respect to the <112> directions of the surface (bottom section in Fig. 2f, g), which are consistent with the angles of $\pm(16 \pm 2)°$ determined in STM images (Fig. 2b).

The calculations reveal a charge reorganization due to the adsorption of DPC on Cu(111) with an overall charge transfer from the surface to the carbene of $0.4\,e^-$ (Fig. 2h), in agreement with the depressions around the carbene center in STM images (Fig. 2c, d). The major charge accumulation is between the carbene center and the closest copper atoms of the surface, indicating a strong interaction between them.

## Chirality flip by vibrational heating

Having identified the enantiomers of DPC, we now explore whether the chirality can be flipped through IET manipulation. On positioning the STM tip above one of the phenyl rings (cross in Fig. 3a), two distinct levels in the $I - t$ traces indicate a change of the DPC molecule between two defined states (Fig. 3b). They correspond to the two enantiomers of DPC as verified by subsequent STM imaging. The high-current level corresponds to the protrusion under the tip (R1, Fig. 3a) and the low-current level to the protrusion moved to the right of the tip (L1, Fig. 3a). The observed conversion of the DPC molecule by IET manipulation is in pairs of $Rn \longleftrightarrow Ln$ ($n = 1, 2$, and $3$), but not the neighboring enantiomers in the angular plot in Fig. 2b, R1↔L3, R2↔L1, or R3↔L2. Thus, the flip of the DPC molecule over the anchoring carbene center is forbidden, the favored mechanism for the chirality change of an unanchored molecule[17]. Here, we determine the immobile carbene center as the anchoring point of the molecule to the surface by performing a bisection of R1 and L1 (Supplementary Fig. 3). It suggests a rotation mechanism of the chirality change around the anchored carbene center. During IET manipulation, the injected electrons initiate the rotation of the two phenyl rings with respect to the carbene center, which changes the adsorption sites of the two rings (Supplementary Fig. 4). It changes the relative heights of the two rings and thereby the chirality of DPC changes. Because of the strong interaction between the carbene center and the surface, neither diffusion nor rotation is observed beyond that what is necessary during the chirality change.

The two clearly distinguishable current levels of the two enantiomers in the $I - t$ traces allow us to quantify the dynamics of the chirality change. For the same values of high and low current, the time intervals between current changes decrease drastically at an increased voltage (Fig. 3b). Based on the time constants $\tau$ from the exponentially distributed probability of the time intervals (Supplementary Fig. 5), we extract the yield of the chirality change per electron (Fig. 3c). The yield changes at three distinct voltages, $(64 \pm 2)$ mV (I), $(125 \pm 2)$ mV (II), and $(161 \pm 1)$ mV (III). A possible mechanism of the chirality change at low energies is the excitation of molecular vibrations. Indeed, the infrared spectra of DPC in rare gas matrices show one skeletal vibrational mode at 62 meV and two C–H deformation modes at 126 meV and 172 meV[23], which fit nicely to the voltages determined here. This correspondence indicates a chirality change of DPC induced by vibrational heating[27]. Such a mechanism is corroborated by $I - V$ curves with symmetric voltage thresholds for current fluctuations in the positive and negative voltage range (Supplementary Fig. 7).

To estimate the number of electrons needed to induce the chirality change, we vary the current at constant voltages (70 mV, 150 mV, and 220 mV) above the three threshold voltages. Power law fittings $\tau^{-1} \propto I^N$ give $N \approx 1$, indicating a one-electron process for the chirality change (Fig. 3d). Therefore, the lowest threshold, at $(64 \pm 2)$ meV, corresponds to an upper value of the energy barrier for the chirality change (see Supplementary Fig. 8).

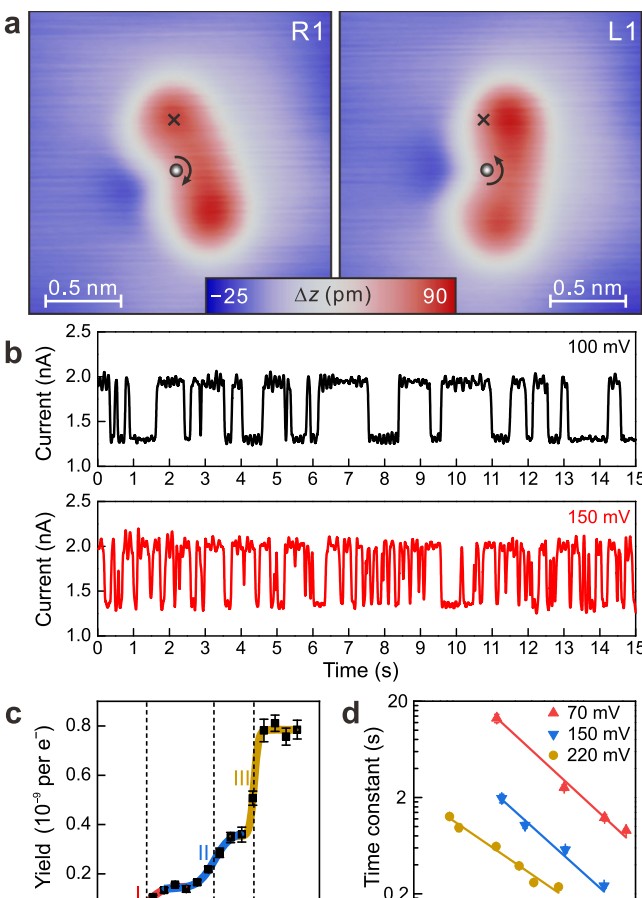

**Fig. 3 | Reversible chirality change of DPC on Cu(111). a** STM images of a DPC molecule before (R1) and after (L1) manipulation. The cross marks the injection site of the inelastic electrons. The gray sphere marks the immobile center of the chirality change (for details see Supplementary Fig. 3). **b** $I - t$ traces recorded at 100 mV and 2 nA (black trace) and 150 mV and 2 nA (red trace) at the cross in (**a**). **c** Chirality change yield $Y$ per electron vs. voltage $V$. Error bars represent the standard deviation by fitting hundreds of switching events. Red, blue, and yellow curves are fits in separated voltage regions based on two-state Boltzmann distributions (for details see Supplementary Fig. 6). Dashed lines mark the threshold voltages at $(64 \pm 2)$ mV (I), $(125 \pm 2)$ mV (II), and $(161 \pm 1)$ mV (III), respectively. **d** Time constant $\tau$ vs. tunneling current $I$ at various voltages. The solid lines are fits to the data by power laws $\tau^{-1} \propto I^N$, yielding $N = 1.25 \pm 0.08$, $1.22 \pm 0.16$, and $0.93 \pm 0.07$ at voltages of 70 mV (red), 150 mV (blue), and 220 mV (yellow), respectively. Scanning parameters: (**a**) $V_b = 10$ mV, $I_t = 1$ nA.

## Enantiomeric excess by weak interactions

Based on this low energy barrier, it is possible to alter the dynamic chirality change of DPC by weak vdW interactions with the STM tip (Fig. 4). Without any external perturbations, the dynamic chirality change of DPC is described by a symmetrical double-well potential, and thus excitation of a single chiral DPC molecule should lead to an equal probability of the formation of each of the enantiomers. However, the occupations in time, $Occ_H$ and $Occ_L$, extracted from a histogram of the $I - t$ traces (Fig. 4a, e, i, m), do not directly represent the respective potential well depths. The occupations in time must be normalized to occupations per electron because, given the same time interval, there are more electrons flowing through the molecule for a higher current value than a lower one. Thus, more switching events will be generated per time interval for a higher current value than a lower one. To compensate for this effect, we define a normalized occupation

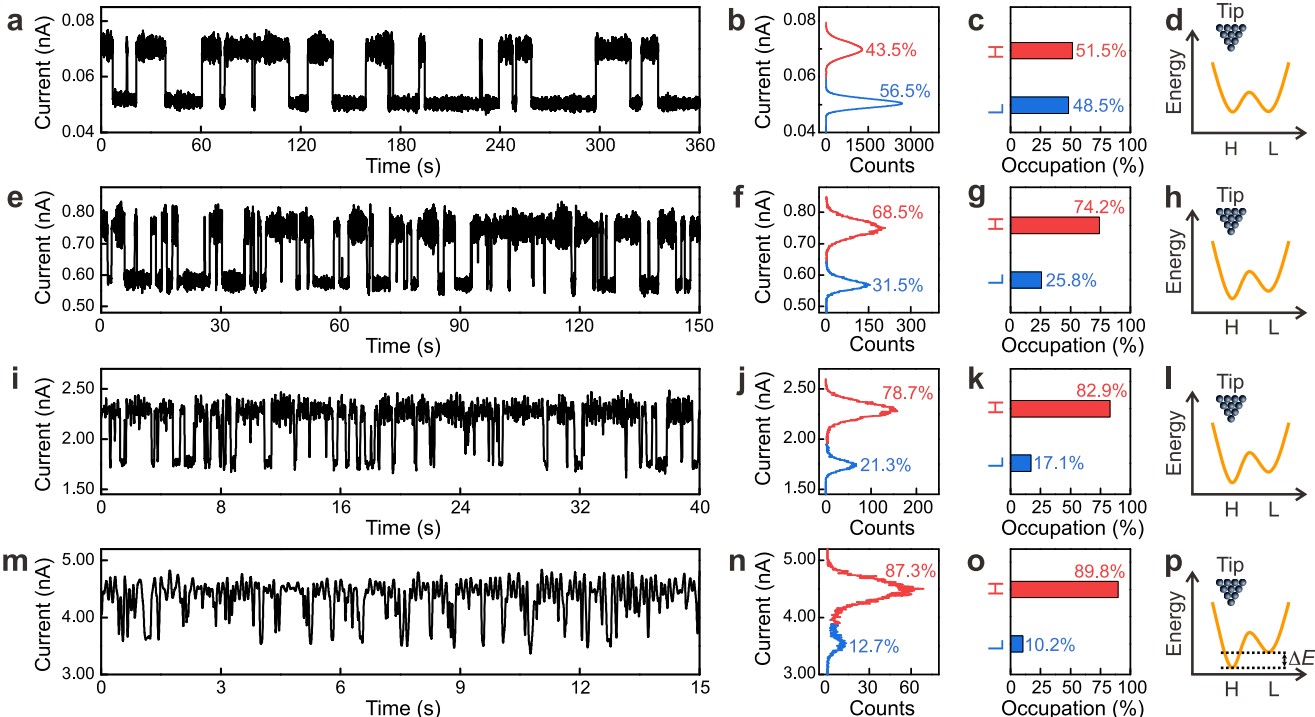

**Fig. 4 | Asymmetric distribution of DPC between its two enantiomers. a–p** IET manipulation at $z$-offsets of 0 nm (**a–d**), 0.14 nm (**e–h**), 0.19 nm (**i–l**), and 0.22 nm (**m–p**) from an initial setpoint (tunneling resistance of 2 GΩ) on one of the phenyl rings. For details of the estimation of the $z$-offsets see Supplementary Fig. 9. From left to right: (**a, e, i, m**) $I − t$ traces, (**b, f, j, n**) current histograms; percentages in panels mark occupations in time for the high-current (red) and low-current (blue) states, (**c, g, k, o**) normalized occupations, and (**d, h, l, p**) schematic double-well potentials (orange curves) in the presence of the STM tip (dark blue spheres). For details about how the normalization was performed see text. $\Delta E$ marks the difference between the two minima of the double-well potential. Note that $\Delta E$ is largely exaggerated with respect to the barrier for the chirality change to make it visible. Setpoint voltages: (**a–d**) 120 mV, (**e–p**) 70 mV.

probability (Fig. 4c, g, k, o)

$$P_i = \frac{\text{Occ}_i \cdot I_i}{\text{Occ}_H \cdot I_H + \text{Occ}_L \cdot I_L} \qquad (1)$$

where $i =$ H or L represent the high-current (H) or low-current (L) states. They reflect the probabilities per electron for DPC staying in the H or L states. For details about how the normalized occupations are inferred from the $I–t$ trace see Supplementary Fig. 10.

The normalized occupations of $P_H : P_L = 51.5\% : 48.5\%$ are almost equal at a relatively large tip-molecule distance (tunneling resistance of 2 GΩ), which is regarded as the case without perturbations (Fig. 4d). Decreasing the tip-molecule distance by a $z$-offset of 0.14 nm increases the normalized occupation $P_H$ under the tip from 51.5 to 74.2% (Fig. 4c, g). At even closer distances, $P_H$ increases to 82.9% ($z$-offset of 0.19 nm, Fig. 4k) and 89.8% ($z$-offset of 0.22 nm, Fig. 4o). This tendency of increasing $P_H$ with decreasing tip-molecule distance is reproducible for other STM tips (Supplementary Fig. 11). The value of $P_H$ slightly varies from tip to tip as expected for different shapes of the tip apex, corroborating further the role of the vdW interactions between the tip apex and the molecule.

The tip could induce the asymmetric distribution of DPC between two enantiomers either by the electric field or vdW interactions. Approaching the tip by a $z$-offset of 0.22 nm (Fig. 4a–m) changes the tip-molecule distance from 0.63 nm to 0.41 nm (for details of the estimation of the tip-molecule distance see Methods), which increases the electric field by a factor of ≈1.5. However, modulating the electric field at a fixed tip-molecule distance ($z$-offset of 0.08 nm) by an even larger factor of four influences the normalized distribution of enantiomers only marginally (Supplementary Fig. 12). Moreover, the $P_H$ and $P_L$ traces at different voltages lead to a perfect overlap (dotted rectangles, Fig. 5a). It

excludes that the electric field induces the asymmetric distribution of the two enantiomers of DPC at closer tip-molecule distances.

To quantify the influence of the vdW interactions with the tip, we extract the energy difference between the two minima in the double-well potential $\Delta E = E_H − E_L$ (Fig. 4p) based on a Boltzmann distribution

$$\Delta E = - kT \ln(P_H / P_L) \qquad (2)$$

where k is the Boltzmann constant, and $T = 5.1$ K is the surface temperature. For details about how $\Delta E$ is inferred from $P_H$ and $P_L$ see Supplementary Fig. 10. $\Delta E$ depends nonlinearly on the $z$-offset with a reduction of 1.23 meV at a $z$-offset of 0.23 nm for $P_H : P_L = 94.5\% : 5.5\%$ (Fig. 5b).

The vdW interactions are approximated by the Lennard-Jones potential

$$E(r) = \epsilon \left[ \left( \frac{r_{min}}{r} \right)^{12} - 2 \left( \frac{r_{min}}{r} \right)^6 \right] \qquad (3)$$

where $r$ is the tip-molecule distance and $r_{min}$ the distance at which the potential energy reaches its minimum at a depth of $\epsilon$. There are two different distances to the DPC molecule in the H and L states, where $r_L = \sqrt{r_H{}^2 + d^2}$ depends on the molecule-molecule distance $d$, determined here as 0.16 nm from the superimposed images of the DPC molecule in its H and L state (inset, Fig. 5b).

Accordingly, fitting the experimentally derived $\Delta E$ by $\Delta E = E(r_H) − E(r_L)$ yields $r_{min} = (0.31 \pm 0.01)$ nm and $\epsilon = (5.0 \pm 0.4)$ meV (black curve in Fig. 5b). $r_{min}$ is consistent with an empirical vdW radius of 0.31 nm estimated by simplifying the phenyl ring and the tip apex as a carbon atom and copper atom, respectively (vdW radii of 0.14 nm and 0.17 nm). The determined $r_{min}$ is smaller than the explored tip-

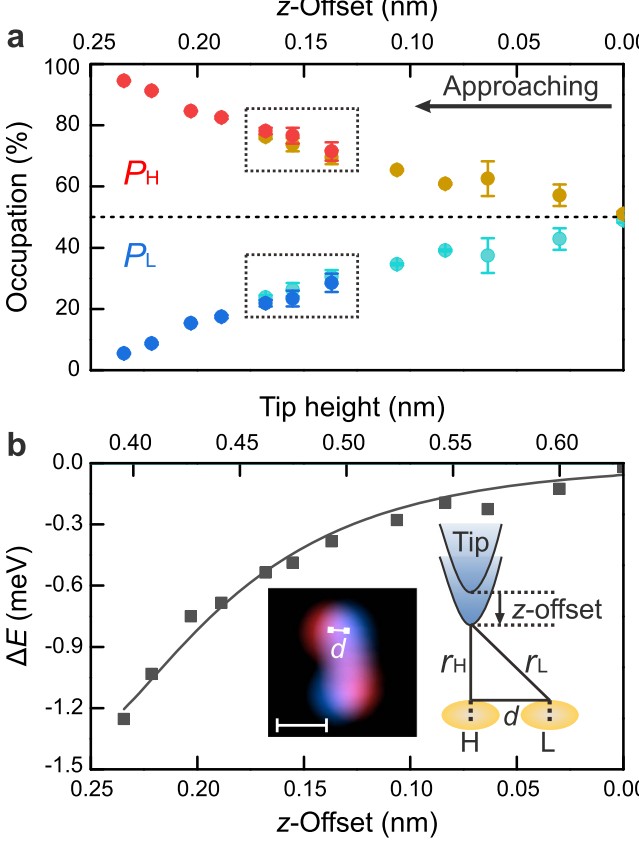

**Fig. 5 | Potential energy difference vs. tip-molecule distance. a** The normalized occupations ($P_H$ and $P_L$) of DPC as a function of the $z$-offsets of the STM tip. Red and blue circles at a voltage of 70 mV; yellow and cyan circles at a voltage of 120 mV. Dotted rectangles mark their overlap. Data acquired using the same tip apex on the same DPC molecule. Error bars represent the standard deviation in two repeated measurements. **b** Potential energy difference ($\Delta E = E_H - E_L$, black squares) derived from $P_H$ and $P_L$ in (**a**). Black curve is a Lennard-Jones potential fitting. Inset: model illustrating the influence of the vdW interactions on the DPC (orange ovals) under the tip (H) and away from the tip (L). The distance $d$ is marked in the superimposed STM images of the DPC molecule in H (red) and L (blue) states. Scale bar: 0.5 nm. Scanning parameters: $V_b = 10$ mV, $I_t = 1$ nA.

molecule distances ($r_H = 0.63 \ldots 0.40$ nm), indicating that the vdW interactions alter the chirality change in its attractive regime. The attractive interactions cause DPC to remain in the enantiomeric state under the tip (Fig. 5a).

Moreover, we employed dynamic AFM to drive the chirality change of DPC, for which only tip-induced interactions were involved without applying bias voltage and inelastic electrons (Supplementary Fig. 13). It is corroborated by AFM manipulation that the chirality change can be induced by tip-induced interactions alone. However, the chirality change in this case requires stronger tip-induced interactions that also induce unintended side processes, such as molecular rotation and translation (Supplementary Fig. 13). Thus, the tip-induced interactions on the chirality change of the molecule cannot be explored by dynamic AFM.

In summary, we present a carbene molecule on a copper surface as a prototype of an anchored molecule with a facile chirality change upon excitation by inelastic electrons. Side processes are not observed under these conditions. Our findings reveal how an enantiomer is stabilized by van der Waals interactions at a single-molecule level. An only few meV change of the potential energy, introduced by weak van der Waals interactions with the tip of a scanning tunneling microscope, is sufficient to induce an asymmetric distribution of the two enantiomers

of this molecule. While addressing the ubiquitous van der Waals interactions between adjacent molecules at the fundamental level remains a grand challenge, our study offers an approach for quantifying its influence on the chirality change of a single molecule. The insight into the change of molecular chirality controlled by van der Waals interactions, an elementary step in chiral induction and amplification, presents a crucial first step to control enantioselective chemical processes precisely. Initialized by the chirality change of a single molecule, the chiral information could be transferred from one molecule to another, leading eventually to homochiral assemblies. While first-principles and empirical calculations started considering weak van der Waals interactions for a more accurate description of molecular behavior[28], our study provides an experimental atomic level approach to study how they substantially alter the molecular dynamics.

## Methods
### Sample preparation
The Cu(111) surface was cleaned by standard cycles of sputtering (Ne$^+$, $3 \times 10^{-5}$ mbar, 1 μA, 1.0 keV, 30 min) and annealing (900 K, 10 min). The carbene precursor DPDM was synthesized according to the procedures as described in Supplementary Note 1. DPDM was sublimed from a glass tube at 270 K, at a pressure of $3.2 \times 10^{-7}$ mbar in a sealed-off molecule deposition chamber (base pressure better than $4 \times 10^{-10}$ mbar). The precursor was exposed from the molecule deposition chamber for 133 s on the bare Cu(111) surface held at 250 K in a preparation chamber. Note that the real pressure at the surface is orders of magnitudes lower than the pressure measured in the molecule deposition chamber.

### STM measurements
STM measurements were performed with a low-temperature STM under ultrahigh vacuum (UHV) conditions (base pressure $< 1.0 \times 10^{-10}$ mbar). All STM images were obtained with a Pt/Ir tip at 5.1 K. The bias voltage was applied to the sample. The STM tip quality was improved by indenting it into the copper surface for a few Å to nm, which leads presumably a tip apex covered by copper atoms. Scanning tunneling spectra were recorded on bare Cu(111) before and after the manipulation to ensure such a metallic tip. Only tips leading to spectra with a clear surface state onset ($\approx -440$ mV)[29] were used for further manipulation.

For IET manipulation, the tip was positioned above a chosen part of the molecule. Then, the current-time ($I - t$) trace was recorded at a chosen bias with the feedback loop off. A steplike change in the tunneling current indicated a successful manipulation, as verified in a subsequent STM image[27,30]. The manipulation was performed at different tip-molecule distances as given by $z$-offsets with respect to the starting set point. Such $z$-offsets were estimated from $I - z$ traces, which were recorded during the vertical approaching of the STM tip to the injection site of the molecule from an initial setpoint at a tunneling resistance of 2 GΩ. The identical forward and backward traces ensured that the STM tips were not altered during the vertical approach. The tip-molecule distance was estimated by approaching the STM tip to touch the surface. The touch was verified by a transfer of the tip-apex atom to the surface in subsequent STM imaging.

The STM topographic images were processed using WSxM[31]. The Laplace-filtered image was processed by a Laplacian matrix:

$$M = \begin{pmatrix} 2 & 2 & 1 & 2 & 2 \\ 2 & 0 & -4 & 0 & 2 \\ 1 & -4 & -12 & -4 & 1 \\ 2 & 0 & -4 & 0 & 2 \\ 2 & 2 & 1 & 2 & 2 \end{pmatrix} \qquad (4)$$

## AFM measurements

Dynamic AFM measurements were performed at 5.1 K in the chamber used for STM measurements. A Kolibri sensor was employed with a resonance frequency of 1 MHz, a spring constant of 540 kN m$^{-1}$, and a quality factor of 18,700[32]. The Kolibri sensor was employed with an electrochemically etched W tip.

To manipulate the molecule by tip-induced interactions, the tip was positioned above a chosen part of the molecule with the STM feedback loop on. Then, the feedback loop was turned off and the bias was decreased to zero. The frequency shift $\Delta f$ was measured during approaching the tip to the molecule. A discrete jump in $\Delta f$ indicated a successful manipulation, as verified in a subsequent STM image.

## Computational details

Gas-phase calculations were performed using Gaussian09 E.01[33]. For the estimation of the barrier between the two chiral states of the molecule, first the structure of the triplet carbene was optimized at the B3LYP-D3/def2-TZVP level of theory[34–37]. Using this starting geometry, the dihedral angle between the two phenyl rings was scanned at the same level of theory by a relaxed PES scan in steps of 1°.

Plane-wave density functional theory calculations were performed with the Quantum-ESPRESSO package, and the charge transfer was calculated using the CRITIC2 software using the output from Quantum-ESPRESSO[38,39]. The surface was simulated using a copper slab composed of four layers of Cu(111) with 6 × 4 atoms in each slab. The bottom two layers were kept fixed at the value of the lattice constant. The PBE functional and D3 dispersion corrections were used for all calculations[36,40]. Ultrasoft pseudopotentials were used with a wavefunction cut-off set to 46 Ry and a charge density cut-off of 8 times the wavefunction cut-off value. Gamma-point only calculations were performed. The adsorption heights of the rings from the carbene with respect to the surface were measured as the difference between the $z$-position of the highest atom for each ring and the average of the $z$-position of the copper atoms from the first layer of the copper slab below the carbene. The slabs were constructed using the atomic simulation environment (ASE)[41].

## Reporting summary

Further information on research design is available in the Nature Portfolio Reporting Summary linked to this article.

# Data availability

The data that support the findings of this study are available from the corresponding author upon request.

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

## Acknowledgements

This work was supported by the Deutsche Forschungsgemeinschaft (DFG, German Research Foundation) under Germany's Excellence Strategy-EXC-2033-390677874 RESOLV, and by the Research Training Group 'Confinement-Controlled Chemistry', which is funded by the Deutsche Forschungsgemeinschaft (DFG, German Research Foundation) under GRK2376//331085229. Y.C. acknowledges the Alexander-von-Humboldt Foundation for a Humboldt Research Fellowship. We acknowledge support by the Open Access Publication Funds of the Ruhr-Universität Bochum.

## Author contributions

Y.C. and K.M. conceived the project. Y.C. performed the STM and dynamic AFM measurements. J.F.R. and W.S. synthesized the precursor molecules and performed the gas-phase calculations. J.M.P. and E.S.G. performed the surface calculations. Y.C. analyzed the data with contributions from the other authors. Y.C. and K.M. wrote the paper with input from all other authors.

## Funding

## Competing interests

The authors declare no competing interests.
