## [Peer Review File · Nature Communications]

Reviewer comments, first round review:

Reviewer #1 (Remarks to the Author):

The paper by Cao et al. describes a careful investigation, at the single-molecule level, of how forces between the tip of a Scanning Tunneling Microscope and an adsorbed molecule can induce the preference of one molecular chiral state over the other. The investigated molecule is conformationally chiral owing to the relative rotations of two phenyl groups which can be distinguished in the STM images. In the adsorbed state the molecule is anchored to the surface by covalent interactions, creating a pivot point about which the molecule can rotate. When electrons from the tip of the STM are injected into the molecule, it undergoes small rotations about the pivot point, leading to rotations of the phenyl groups and thereby switching of the molecular chirality. When the tip is present over one of the phenyl groups, this group prefers a "high state" closer to the tip, creating a preference for one chiral conformational state. By careful experiments, the authors elucidate the kinetics of the current--induced single electron switching process and show that the chiral preference is not caused by the electric field from the STM tip, but rather by the attractive forces between tip and molecule (VdW as modelled by a Lennard-Jones potential).

While many previous studies exist of current-induced molecular conformational switching and rotation, the present papers investigation of interaction forces and the coupling to chiral switching in my view makes it novel and highly interesting. Molecular chiral assemblies have been widely studied (as pointed out by the authors) and chiral induction processes are controlled by molecular interactions. The present paper provides detailed quantitative insight into such interactions (in the form of molecule-tip atom interactions in the present case), making it an important contribution to the detailed understanding of how chiral assemblies form, which has wide impact e.g. regarding the origins of biomolecular homochirality. As such I recommend the paper for publication in Nature Communications.

The paper is well presented and based on careful STM experiments and supporting DFT modeling. In two places I found the paper hard to follow and would encourage some revision to make these parts clearer and more well argued:

(i) The discussion of chirality change yield in relation to Fig. 2C. Looking at the curves of this figure, I would say that the yield stabilizes at a plateau when the mentioned thresholds (arrows) are approached from the low-voltage side. This appears in contrast to the authors argument that new vibrational excitations become possible when the thresholds are crossed, since one would then expect the yields to increase when new vibrational excitations become possible by crossing the threshold voltages (the correspondence with the IR-measured vibrational modes for the molecule is indeed striking). Possibly, the described situation is an artefact of the logarithmic scale used in Fig. 2C and how the threshold voltage $V_{1/2}$ is represented on this scale. To clarity, I would recommend a more detailed discussion of this aspect, in particular the choice of the Boltzmann model mentioned in the caption of Fig. 2c and the fitting to this function. Perhaps one could show one of the curve sections on a linear scale in the SI.

(ii) The discussion of the how residence times in the two states can be translated to the potential well depths (page 10) is not entirely clear to me, particularly I would encourage better arguments for the choice of the used normalized occupation function.

Minor correction: In the caption of Fig. 4a, "blue and purple" should be changed to "green and purple" to be consistent with the figure.

Reviewer #2 (Remarks to the Author):

The manuscript by Cao et al. reports on the chirality change of a single molecule realized with the

tip of a scanning tunnelling microscope. The authors present diphenylcarbene as a prototypical molecule for chirality change observation. A covalently bound molecule is a convenient system to study the chirality change since it cannot be manipulated, nor rotated. The chirality change can be detected in the change of the tunnelling current as the two phenyl rings do not have the same height.

In general, this manuscript is a carefully executed study with a clearly structured interpretation, but it is hard to see the novelty and ground-breaking character. Also, given the interpretation by the influence of vdW, it would seem natural to employ a more advanced methodology, i.e. dynamic AFM spectroscopy, which enables evaluation and quantification of the vdW and electrostatic contributions.

I have several questions and comments which should be addressed for considering the manuscript for publication:

- 1) It is mentioned that this study paves the way to construct homochiral mol. assemblies in solutions and on solid surfaces, however, no hint is given as to how it should be realised, leaving the reader in doubts. Please elaborate on that or tone down these claims.
- 2) I would appreciate a clearer statement of significance, in particular, what is the added value of this study compared to the previous works in the field. This is essential for the broad readership of Nature Communications not very familiar with the specialized field of "on-surface chirality".
- 3) As already mentioned above, I do not see a direct link between the study of a single molecule and the understanding of "chiral induction and amplification". In this case, the molecule is switched with inelastic electrons when a low barrier is overcome. This is not the case for homochiral growth, as there is no tip.
- 4) Can the molecule be switched with zero bias by a vdW force only (either attractive or repulsive)?
- 5) In Fig. 3 there are sketches of the double-well potential in the presence of the STM tip. Can the sketches be plotted quantitatively based on the fitting in Fig. 4? To me, the ΔE that induces a big difference in the normalized occupations is surprisingly small (less than 1.2 meV). Can the ΔE be interpreted as a difference between the two minima of the double well? If yes, can you estimate the barrier height? I would expect it to be in a similar order of magnitude as the ΔE .
- 6) While the introduction and the beginning of the manuscript are soundly described, the last two paragraphs are a bit sparse in my opinion. I am missing a short discussion with outlook, i.e. how it can be useful for the understanding of the key problems in the chirality induction and amplification. Can the obtained values serve as an input for e.g. Monte Carlo simulations of a homochiral growth? Or is the lowering of the barrier on the surface compared to the gas phase the noteworthy result?

Minor comments

On page 9 the expression "constant current" is used. This is a bit misleading since the I-t traces were measured at a constant height.

How does the molecule behave at negative sample biases?

Reviewer #3 (Remarks to the Author):

The manuscript written by Cao et al. demonstrated chiral change of DPC molecules adsorbed on copper surface induced by inelastic tunneling electrons. The carbene center covalently bounds to the copper surface as an anchor, suppressing other side-processes upon excitation. Furthermore, through numerical analysis based on the Lennard-Jones potential, the critical role of vdW

interactions in inducing chiral changes was revealed. Overall, I think this is a nice paper. The results are interesting and significant. The experimental data are of high quality with systematic data analysis. I would recommend its publication in Nature Communications after the authors addressing the following issues.

1. There are DFT calculations presented in Fig. 1 and the energy barrier for the chirality change is presented in Fig. S1 for the free molecule. The authors mentioned such barrier decreases upon adsorption on surface, and is also influenced by the vdW interaction between the tip and the molecule. Those experimental observations should in principle be desirably substantiated from DFT calculations. The authors should provide such DFT results to compare with their experiments.
2. The authors mentioned the data shown in Fig. 3 are reproducible with other STM tips. I recommend to show the data with another tip in Supplementary materials. This is also useful to confirm the role of the vdW interaction between the tip and the molecule, whose magnitude could depend on the actual shape of the tip apex.
3. In page 13 of the main text, the authors estimated the tip-molecule interaction is in the attractive regime based on a simplification of the tip as a copper atom and the molecule as a carbon atom. The authors should add some discussions to justify such a simplification.
4. The red color shown in Figure 1(h) is close to the Cu atoms. It is better to change it to a different color.

Reply to Reviewer for manuscript NCOMMS-22-46174-T

We thank the reviewers for the constructive comments and suggestions on how to improve our manuscript. We have addressed these comments point-by-point and revised the manuscript accordingly.

Comments in *black italic* - Replies in regular black - Amendments to the manuscript in blue.

Reply to reviewer #1:

We thank this reviewer to name our study “*a careful investigation*” and acknowledge that we performed “*careful experiments*”. The reviewer states that the “*investigation of interaction forces and the coupling to chiral switching in my view makes it novel and highly interesting*”, and that it “*provides detailed quantitative insight...making (it) an important contribution to the detailed understanding of how chiral assemblies form, which has wide impact...*”. We highly appreciate the reviewer’s positive comments to our work and his/her recommendation for publication in *Nature Communications*. We carefully addressed the open points raised by the reviewer below.

1. The discussion of chirality change yield in relation to Fig. 2C. Looking at the curves of this figure, I would say that the yield _stabilizes_ at a plateau when the mentioned thresholds (arrows) are approached from the low-voltage side. This appears in contrast to the authors argument that new vibrational excitations become possible when the thresholds are crossed, since one would then expect the yields to _increase_ when new vibrational excitations become possible by crossing the threshold voltages (the correspondence with the IR-measured vibrational modes for the molecule is indeed striking). Possibly, the described situation is an artefact of the logarithmic scale used in Fig. 2C and how the threshold voltage $V_{1/2}$ is represented on this scale. To clarity, I would recommend a more detailed discussion of this aspect, in particular the choice of the Boltzmann model mentioned in the caption of Fig. 2c and the fitting to this function. Perhaps one could show one of the curve sections on a linear scale in the SI.

Reply: We agree with the reviewer that the logarithmic scale is misleading. In the revised manuscript, we use a linear scale as suggested (new Fig. 3c, former Fig. 2c). In addition, we replaced the short arrows by dashed lines to mark the thresholds more clearly.

The three thresholds are determined by one Boltzmann function for every threshold between two yield levels. We added Supplementary Fig. 6 with a detailed description

of the Boltzmann function. The threshold is given by the voltage of the point in the middle between two levels, i.e., the voltage $V_{1/2}$ at $Y_{1/2} = (Y_{low} + Y_{high})/2$ (Fig. R1).

Fig. R1 (Supplementary Fig. 6). Determination of voltage threshold between two yield levels: Chirality change yield Y per electron vs. bias voltage V . Black squares: experimental data; yellow line: result of fitting a Boltzmann function; black lines: two fitted yield levels Y_{low} and Y_{high} ; blue sphere: the middle point at $Y_{1/2} = (Y_{low} + Y_{high})/2$.

2. The discussion of the how residence times in the two states can be translated to the potential well depths (page 10) is not entirely clear to me, particularly I would encourage better arguments for the choice of the used normalized occupation function.

Reply:

Fig. R2 (Supplementary Fig. 10). Procedure to derive potential well depths from $I-t$ trace via the occupations in time (Occ.) and the normalized occupation (Normd. Occ.).

We are sorry that our description was unclear. We added a description to the revised Supplementary Information (Supplementary Fig. 10) and refer the reader to it in the main text. In the revised version, we renamed the “residence times” as “occupations in time” since the frequency counts of the $I-t$ trace are not in a time unit.

It is demonstrated for one set of data (see Supplementary Fig. 10) corresponding to the data presented in Fig. 4e-h. There are three steps to convert the $I-t$ trace via the occupations in time and the normalized occupations to the potential well depths (Fig. R2).

Step I: We extract the occupations in time, Occ_H and Occ_L from a histogram of the time trace ($I-t$ trace). The occupations display two well-separated maxima at the current values I_H and I_L . Their relative areas yield the percentages Occ_H and Occ_L during which the molecule is in its high state (H) and its low state (L).

Step II: The occupations in time must be normalized to occupations per electron because, given the same time interval, there are more electrons flowing through the molecule for a higher current value than a lower one. Thus, more switching events will be generated per time interval for a higher current value than a lower one. To compensate for this effect, we define a normalized occupation probability $P_i = (Occ_i \cdot I_i) / (Occ_H \cdot I_H + Occ_L \cdot I_L)$, where I_i with $i = H$ or L represent the high-current (H) or low-current (L) states, respectively. They reflect the probabilities per electron for the molecule staying in H or L states.

Step III: The molecule switching between the two states is considered as a classic system following a Boltzmann distribution. Thus, the energy difference is extracted by the Boltzmann distribution $P_H/P_L = e^{-\Delta E/kT}$, where $\Delta E = E_H - E_L$ is the energy difference between the two minima of the double-well potential with a depth of E_H and E_L , k is the Boltzmann constant, and T is the surface temperature. For the example in Fig. R2, ΔE is calculated as 0.5 meV from $P_H : P_L = 74.2\% : 25.8\%$ at $T = 5.1$ K.

Minor correction:

In the caption of Fig. 4a, “blue and purple” should be changed to “green and purple” to be consistent with the figure.

Reply: We corrected it. During the preparation of the revised manuscript, we realized that the color scheme “red and green” is not recommended according to the formatting requirements of *Nature Communications*. Therefore, we use another color scheme “red and blue” to represent the high and low current states throughout the manuscript.

Reply to reviewer #2:

We thank the reviewer for naming our work “*a carefully executed study with a clearly structured interpretation*”. We also thank the reviewer for the constructive comments and suggestions on how to improve our manuscript. We carefully addressed each point as detailed below.

The reviewer remarks that “*it would seem natural to employ a more advanced methodology, i.e. dynamic AFM spectroscopy, which enables evaluation and quantification of the vdW and electrostatic contributions*”.

Reply:

Fig. R3 (Fig. 1). Schematics of molecular chirality change induced by adjacent molecule or tip. (a) Schematic potential wells for chirality changes. Solid gray curve represents the native potential well for the chirality change of a molecule without interaction; dashed orange curve represents the net potential induced by an adjacent molecule or tip, and solid orange curve represents the total potential well in the presence of an adjacent molecule or tip. E_a marks the activation energy of a chirality change. e^- and ΔT mark inelastic electrons injected by a STM tip and temperature, respectively. (b) Schematic chirality changes of a single molecule without (w/o) interaction and with (w/) interaction; dashed arrows mark molecule-molecule interactions and tip-molecule interactions.

We thank for the suggestion and address this concern from two approaches:

- We agree with the reviewer that AFM is, generally, a very powerful tool for interaction measurements. However, conventional AFM studies usually employ the strong forces in the repulsive regime for manipulation (e.g., Nat. Chem. 8, 935 (2016), J. Am. Chem. Soc. 142, 10673 (2020), added to the manuscript as Ref. 12, 13). In contrast, our study involves the weak vdW interactions in the attractive regime. To mimic a chiral induction process in a molecular assembly, we proposed a strategy which does not rely purely on forces but combines the effect of inelastic electrons with tip-induced vdW interactions (see Fig. R3). The inelastic electrons injected into the molecules serve to overcome the energy barrier of the chirality

change replacing the temperature effects in the molecular assembly. The tip-induced vdW interactions represent the intermolecular vdW interactions, altering the potential wells during chiral induction. Such a strategy employs the inelastic electrons instead of tip-induced interactions as a driving force in chiral induction, making it possible to address the influence of weak vdW interactions in the dynamics of chirality changes. In reply to the reviewer's comment, we added the schematic Fig. R3 (new Fig. 1) to the revised manuscript, and extended the introduction to enforce this aspect.

Fig. R4 (Supplementary Fig. 13). Manipulation of DPC by tip-induced interactions only. (a-e, g-i) STM images of sequential manipulation of two DPC molecules ((a-e):I, (g-i): II) by tip-induced interactions. The manipulation sites in (a-d) are the brighter protrusions and in (g,h) the less bright protrusions, as marked by crosses in (a) and (g), respectively. Contours (solid) in (a) and (g) mark the initial positions of DPC; contours (dotted) in (b-e) and (h,i) mark the positions of DPC in the previous frame. Scanning parameters: $V_b = 10$ mV, $I_t = 10$ pA. (f) Spectra of frequency shift Δf as a function of z-offset measured during the manipulation from (a) to (b). The z-offset is defined with respect to an initial setpoint of $V_b = 10$ mV and $I_t = 10$ pA. The blue arrow marks the approaching direction. The red arrow marks a discrete jump of Δf during the manipulation.

- Moreover, to comply with the reviewer's suggestion, we employed dynamic AFM spectroscopy to characterize the tip-induced interactions during the chirality switch at zero bias. As shown in Fig. R4f, it demands a much larger z-offset. The

strong tip-induced interactions change not only the chirality of the molecule, but also induce unintended side processes, such as molecular rotation (Fig. R4a to R4e) and translation (Fig. R4g to R4i). These side processes undermine characterizing the influence of tip-induced interactions on the chirality change of the molecule by dynamic AFM. The experiment demonstrates that AFM spectroscopy is not suitable to characterize the weak vdW interactions in the chirality change of the molecule. In the revised version, we added Supplementary Fig. 13 and refer the reader to it in the main text (pages 14,15).

Apart from this general concern, the reviewer has several questions and comments which we have addressed as follows:

1. It is mentioned that this study paves the way to construct homochiral mol. assemblies in solutions and on solid surfaces, however, no hint is given as to how it should be realised, leaving the reader in doubts. Please elaborate on that or tone down these claims.

Reply: We agree with the reviewer that this aspect was not clearly stated in the original version of the manuscript. To construct homochiral molecular assemblies in solutions and on solid surfaces, an elementary step is chiral induction, i.e., a molecule prefers one enantiomer in favor of the other when it is adjacent to a chiral molecule. The chiral induction initiates the chiral flip of the molecules, leading eventually to homochiral assemblies. While chiral induction is dominated by intermolecular interactions like vdW interactions and hydrogen bonding, the characterization of these interactions at the fundamental level is still missing. Our study provides a strategy to address this issue by using tip-induced vdW interactions to simulate the ones induced by the molecule. We reveal how an enantiomer is stabilized by vdW interactions at a single-molecule level. The unprecedented control of molecular chirality by vdW interactions is fundamental for chiral induction and amplification mechanisms. However, we agree with the reviewer that to realize a construction of assemblies in solutions based on our fundamental study is not immediate.

In the revised manuscript, we have toned down the last sentence in the abstract to “Our model study provides an unprecedented view into the change of molecular chirality controlled by van der Waals interactions, which is fundamental for understanding the mechanisms of chiral induction and amplification”.

To support this statement, we added the new Fig. 1 (see Fig. R3 above) and extended the introduction to describe our strategy. In addition, we echoed this part in the conclusion with more discussions about the link between our study of a single molecule and the general understanding of chiral induction and amplification.

2. I would appreciate a clearer statement of significance, in particular, what is the added value of this study compared to the previous works in the field. This is essential for the broad readership of Nature Communications not very familiar with the specialized field of “on-surface chirality”.

Reply: We thank the reviewer for this suggestion. Previous work in the field reported chirality changes of molecules upon excitation by photons (Ref. 14), heat (Ref. 15), and electrons (Refs. 16,17,19). However, these studies employed excitations with higher energies, yielding random chirality changes but missing control. These studies deepened the fundamental understanding of molecular chirality, but are not related to the chiral induction in nature, which is altered by weak intermolecular interactions. In our study, the subtle influence of vdW interactions is used to gain unprecedented control over the chirality changes. To achieve this, we:

- choose a prototypical anchored molecules with a low energy barrier to switch between its two enantiomeric states.
- propose a STM-based strategy which combines the effect of inelastic electrons with tip-induced vdW interactions. The former one serves to overcome the energy barrier, while the latter alters the potential wells of the chirality change.

With this approach, we successfully mimic a chiral induction process in a molecular assembly.

To better position our work in the context of the existing literature,

- we added to the introduction “Overall, though chirality changes within individual molecules were achieved by photons (Ref. 14), heat (Ref. 15), and electrons (Refs. 16,17,19), the key to chiral induction, i.e., the influence of vdW interactions on this process, has not been addressed yet.” (page 4).
- we wrote at the end of the introduction “Our study takes advantage of inelastic electrons and tip-induced vdW interactions to mimic a chiral induction in a molecular assembly. The subtle influence of vdW interactions is used to gain control over the chirality, providing a molecular-level understanding of chiral induction and control.” (page 4).

3. As already mentioned above, I do not see a direct link between the study of a single molecule and the understanding of “chiral induction and amplification”. In this case, the molecule is switched with inelastic electrons when a low barrier is overcome. This is not the case for homochiral growth, as there is no tip.

Reply: As detailed in the answer to remark 1, we added a scheme as new Fig. 1 to clarify this link. In brief, it is part of our strategy to use inelastic electrons to alter the chirality of the molecule, which mimics the temperature for overcoming the energy barrier in homochiral induction. Such a strategy employing the inelastic electrons

instead of tip-induced interactions as a driving force in chiral induction allows to address the role of weak vdW interactions in the dynamics of chirality changes.

4. *Can the molecule be switched with zero bias by a vdW force only (either attractive or repulsive)?*

Reply: Yes, it is possible to switch the molecule by tip-induced forces only (new Supplementary Fig. 13). However, as detailed in the answer to the general remark above, the strong tip-induced interactions change not only the chirality of the molecule, but also induce undesired side processes, such as rotation and translation.

5. *In Fig. 3 there are sketches of the double-well potential in the presence of the STM tip. Can the sketches be plotted quantitatively based on the fitting in Fig. 4? To me, the ΔE that induces a big difference in the normalized occupations is surprisingly small (less than 1.2 meV). Can the ΔE be interpreted as a difference between the two minima of the double well? If yes, can you estimate the barrier height? I would expect it to be in a similar order of magnitude as the ΔE .*

Reply: Indeed, ΔE represents the difference between the two minima of the double-well potential (see revised Fig. 4p, former Fig. 3d-iv).

We can estimate the upper limit of the barrier height from the current experiment. based on the vibrational excitation yield (Fig. 3c, former Fig. 2c) and that it is a one-electron process (Fig. 3d, former Fig. 2d). The upper limit of the barrier height is 64 meV (Supplementary Fig. 8). In the revised manuscript, we added a discussion about estimating the upper limit of the barrier height (page 10).

The barrier height is almost two orders of magnitude larger than ΔE . Therefore, a scaled plot of the double-well potential makes ΔE invisible, even for its largest value. For this reason, the changes are sketched qualitatively to illustrate that the potential well deepens in the presence of the STM tip.

In the revised manuscript (page 14), we mark \$\Delta E\$ in Fig. 4p and point out in the main text “the energy difference between the two minima in the double potential well \$\Delta E = E_H - E_L\$ (Fig. 4p)”.

6. *While the introduction and the beginning of the manuscript are soundly described, the last two paragraphs are a bit sparse in my opinion. I am missing a short discussion with outlook, i.e. how it can be useful for the understanding of the key problems in the chirality induction and amplification. Can the obtained values serve as an input for e.g. Monte Carlo simulations of a homochiral growth? Or is the lowering of the barrier on the surface compared to the gas phase the noteworthy result?*

Reply: We thank the reviewer for the questions. The obtained values cannot serve as an input for Monte Carlo simulations because Monte Carlo simulations depend on energy barriers, for which we can only give an upper bound as pointed out in the reply to the previous question. The main results are quantifying the influence of vdW interactions on the chirality change on a single-molecule level, which is challenging to be validated experimentally. For which reason understanding vdW interactions is strongly theory-based at present. We consider the lowering of the barrier on the surface compared to that in the gas phase as of minor importance.

We extended the conclusion with an outlook from three approaches:

- We point out that our study offers a novel approach in terms of characterizing the role of weak vdW interactions in chiral induction:
“While addressing the ubiquitous van der Waals interactions between adjacent molecules at the fundamental level remains a grand challenge, our study offers a novel approach for quantifying its influence on the chirality change of a single molecule.”
- We explained the link between our study of a single molecule and general understanding of chiral induction and amplification:
“The unprecedented view into the change of molecular chirality controlled by van der Waals interactions, an elementary step in chiral induction and amplification, presents a crucial first step to control enantioselective chemical processes precisely. Initialized by the chirality change of a single molecule, the chiral information could be transferred from one molecule to another, leading eventually to homochiral assemblies.”
- Finally, we suggest more considerations of vdW interactions in dynamic theory:
“While first-principles and empirical calculations started considering weak van der Waals interactions for a more accurate description of molecular behavior (Ref. 28), our study provides an unprecedented experimental atomic level approach to study how they substantially alter the molecular dynamics.”

Minor comments:

On page 9 the expression “constant current” is used. This is a bit misleading since the I-t traces were measured at a constant height.

Reply: We agree with the reviewer that the expression is misleading. We changed it to “for the same values of high and low current” (page 10 of the revised manuscript).

How does the molecule behave at negative sample biases?

Reply: At negative biases, the molecule behaves similarly as at positive biases. We added Supplementary Fig. 7 in the revised Supplementary Information for illustration. The I - V curve exhibits current fluctuations at similar values for both negative and positive biases (Fig. R5). Note that the nearly symmetric onset of current fluctuations in the I - V curve supports the vibrational heating mechanism of the chirality change. For this reason, we concentrated on the positive voltage range to study the chirality change and the enantiomeric excess by weak interactions. Yet, the conclusions also apply to the negative regime.

Fig. R5 (Supplementary Fig. 7). I - V curve recorded at the crosses in the STM image in corresponding colors. The bias was ramped at a step size of 2 mV and a dwelling time of 2 s for each step. Gray background marks range of fluctuations above the molecule (black curve). Inset: STM image of a DPC molecule recorded at scanning parameters of $V_b = 10$ mV, $I_t = 10$ pA.

Reply to reviewer #3:

We thank this reviewer to name our study “*interesting and significant*” and remark that “*experimental data are of high quality with systematic data analysis*”. We highly appreciate the reviewer for his/her valuable comments and the support for the publication of our work in *Nature Communications* after revision. We have clarified all the open questions below. Please note that former Fig. 1 was renamed Fig. 2 and former Fig. 3 was renamed Fig. 4 in the revised manuscript because a new Fig. 1 was added.

1. There are DFT calculations presented in Fig. 1 and the energy barrier for the chirality change is presented in Fig. S1 for the free molecule. The authors mentioned such barrier decreases upon adsorption on surface, and is also influenced by the vdW interaction between the tip and the molecule. Those experimental observations

should in principle be desirably substantiated from DFT calculations. The authors should provide such DFT results to compare with their experiments.

Reply: We agree with the reviewer that the calculated barrier would be a desirable addition to the manuscript. However, we note that, while the calculation of energy barriers in gas phase is relatively straightforward and there are plenty of well-established algorithms developed to that end, energy barriers on the surface are in practice difficult to converge. We indeed attempted to calculate the barrier for the chirality change. However, the structure of the transition state (TS) for the transformation between both isomers on the surface did not reach convergence. We note that, apart from the dihedral rotation that controls the barrier in gas phase, the process is much more complex on the surface since it also involves the changing of the adsorption sites for the phenyl rings of the molecule. We applied state-of-the-art strategies to calculate energy barriers on surfaces: the Nudged Elastic Band (NEB) method for calculating the minimum energy path and the climbing image-NEB (CI-NEB) method for optimizing possible TS structures. The latter was also employed to scan the potential energy surface in order to locate the TS connecting the initial and final isomers. Unfortunately, none of these strategies led to the identification and optimization of the TS. Therefore, this is not included in the manuscript. Nevertheless, we would like to note that a probable reason of why the transition state search did not result in a converged structure would be a relatively flat potential energy surface describing the transformation between both isomers. However, without a calculated value for the energy barrier, and given the intrinsic challenges of such calculations on the surface, we refrain from making a final statement on that hypothesis based on calculations. However, based on our experiment, the energy barrier must be lower than 64 meV (Supplementary Fig. 8) and thus reduced from the gas phase value of 72 meV (Supplementary Fig. 1).

Nevertheless, we consider the lowering of the energy barrier on the surface compared to that in the gas phase as of minor importance, which will not influence the conclusions of this manuscript. Therefore, in the revised manuscript (page 10), we removed the statement “Thus, upon adsorption on the surface, the energy barrier of the chirality change of DPC is even reduced from its gas phase value of 72 meV (Supplementary Fig. 1).”.

2. The authors mentioned the data shown in Fig. 3 are reproducible with other STM tips. I recommend to show the data with another tip in Supplementary materials. This is also useful to confirm the role of the vdW interaction between the tip and the molecule, whose magnitude could depend on the actual shape of the tip apex.

Reply: We added another set of data recorded with another tip apex in Supplementary Fig. 11 of the revised Supplementary Information. It confirms the trend of P_H increasing with decreasing tip-molecule distance (Fig. R6). Though the orders of magnitude are comparable, P_H slightly varies from tip to tip as expected for different shapes of the tip apices, for example, it is 89.8% for tip #1 at a z-offset of 0.22 nm but 96.8% for tip #2 at a smaller z-offset of 0.21 nm. Their slight variations corroborate further the role of the vdW interactions between the tip apex and the molecule.

Fig. R6 (Supplementary Fig. 11). Tip-induced enantiomeric excess. IET manipulation on one of the phenyl rings of DPC at a z-offset of 0.07 nm (a-c), 0.14 nm (d-f), 0.20 nm (g-i), and 0.21 nm (j-l) from an initial setpoint at tunneling resistance of 2 G Ω . From left to right: (a,d,g,j) $I-t$ traces, (b,e,h,k) current histograms; percentages in panels mark occupations in time for the high-current (red) and low-current (blue) states, (c,f,i,l) normalized occupations (Normd. Occ.). Setpoint voltages: (a,b) 150 mV, (c) 50 mV, (d) 40 mV.

3. In page 13 of the main text, the authors estimated the tip-molecule interaction is in the attractive regime based on a simplification of the tip as a copper atom and the molecule as a carbon atom. The authors should add some discussions to justify such a simplification.

Reply: We apologize that our text was misleading. In fact, we did not estimate the tip-molecule interactions based on such a simplification. We estimated the tip-molecule interactions to be in the attractive regime because the explored tip-molecule distances are at $r_H = 0.63 \dots 0.40$ nm larger than $r_{\min} = 0.31$ nm, the value obtained by fitting the data with the Lennard-Jones potential. The simplification of the tip and the

molecule as a copper atom and carbon atom is only used to compare an empirical vdW radii to the fitted r_{\min} . Their correspondence indicates that the fitting yields a reasonable value. In the revised manuscript, we reformulated the sentence to clarify this issue (page 14).

4. The red color shown in Figure 1(h) is close to the Cu atoms. It is better to change it to a different color.

Reply: We agree with the reviewer that the red color is too close to the Cu atoms. We changed it to pink in the revised manuscript (Fig. 2h, former Fig. 1h).

Reviewer comments, further round review:

Reviewer #1 (Remarks to the Author):

I have with great interest read the revised version of the manuscript by Morgenstern and co-workers on STM-tip induced molecular chirality changes through VdW interactions. The revised Fig. 3c and the new SI Fig. 6, along with the accompanying text, very well addresses the two questions I raised in my report regarding more detailed explanations for various aspects of the data analysis, Furthermore I find the authors have succeeded in more clearly elaborating the significance of their work by the newly added Fig. 1 and the description of how the STM-induced VdW interactions should be viewed as a controllable mimic of the VdW interactions ubiquitous in molecular self-assembly. The authors should also be complemented for adding measurements performed by dynamic NC-AFM, while at the same time arguing how their STM-based approach is more suitable for addressing the considerably weaker VdW interactions in the non-contact regime. Overall, I thus find the authors have thoroughly addressed all comments raised in the three referee reports, resulting in significant improvements on an already very good piece of work, which I recommend for publication in its present form.

Reviewer #2 (Remarks to the Author):

The authors answered all my points. They even performed an additional dynamic AFM study, which I suggested, and clarified why STM is more suitable for studying weak van der Waals interactions. I recommend the publication of the revised manuscript in Nature Communications.